# Ultrasound Evaluation of Therapeutic Response to Tofacitinib and Upadacitinib in Patients with Rheumatoid Arthritis—Real-Life Clinical Data

**DOI:** 10.3390/biomedicines13061339

**Published:** 2025-05-30

**Authors:** Vladimira Boyadzhieva, Konstantin Tachkov, Soner Emin, Zhaklin Apostolova, Rumen Stoilov, Guenka Petrova, Nikolay Stoilov

**Affiliations:** 1Department of Rheumatology, University Hospital “St. Iv. Rilski”, Medical University of Sofia, 1612 Sofia, Bulgaria; dr.soneremin@gmail.com (S.E.); rmstoilov@abv.bg (R.S.); dr_nstoilov@yahoo.com (N.S.); 2Faculty of Pharmacy, Medical University of Sofia, 1000 Sofia, Bulgaria; ktashkov@pharmfac.mu-sofia.bg (K.T.); gpetrova@pharmfac.mu-sofia.bg (G.P.); 3First Department of Internal Medicine, Rheumatology Clinic, UMBAL “St. Marina”—Varna, Medical University of Varna, 9010 Varna, Bulgaria; jackieapostolova@gmail.com; 4Research Institute of Innovative Medical Sciences “InnoMedSci”, 1431 Sofia, Bulgaria

**Keywords:** tofacitinib, upadacitinib, musculoskeletal ultrasonography, rheumatoid arthritis, disease activity

## Abstract

**Background**: In recent years, musculoskeletal ultrasonography (MSUS) has established itself as a reliable method for evaluating disease activity in combination with clinical examination and laboratory tests. **Objectives**: In this pilot study, we aimed to evaluate the ultrasound response to treatment with tofacitinib and upadacitinib on tendons and joints in comparison to clinical and laboratory results in patients with RA who have shown inadequate response to conventional synthetic and/or biologic disease-modifying antirheumatic drugs (cs/b DMARDs). **Methods**: This study presents the MSUS assessment of therapeutic response in RA patients treated with tofacitinib or upadacitinib over a 24-week period. In a prospective, single-center study, patients were treated with upadacitinib 15 mg/daily or tofacitinib 2 × 5 mg/daily or 11 mg/daily, in combination with or without methotrexate or another conventional DMARDs. Disease activity was assessed by DAS28-CRP, CDAI, and SDAI, as well as MSUS. Patients were evaluated at baseline for ultrasound measures and at weeks 2, 4, 8, 12, and 24 for the rest of the indicators. For each patient, we used two ultrasound (US) scores (gray-scale, GS, and power Doppler, PD scores) and the system of European Alliance of Associations for Rheumatology outcome measures in rheumatology (EULAR-OMERACT) US scoring (combined GS and PD). We also calculated the tenosynovitis score (GS and PD) according to OMERACT recommendations. **Results**: A total of 53 patients were recruited. A total of 25 patients with a mean age of 56 ± 11.6 SD were followed in the upadacitinib group, and 22 patients with a mean age of 56.9 ± 11.3 were followed in the tofacitinib group. At baseline, DAS28-CRP for the upadacitinib group was 5.57 ± 1.24, and for tofacitinib, it was 4.77 ± 1.47. The baseline visit (GS, PD, and combined—US) and tendon US scores (GS and PD) were, respectively, 23 ± 2.96, 15 ± 2.56, 24.08 ± 3.36, 11.04 ± 2.21, and 8.44 ± 1.65 for the upadacitinib group. USGS-J—23 ± 3.55, USPD-J—13.36 ± 2.44, OMERACT composite—23.4 ± 3.84, USGS-T—12.18 ± 2.23, and USPD-T—9.5 ± 1.92 were found in the patients treated with tofacitinib. In both groups of patients, a significant reduction was found in both DAS28-CRP and the described MSUS scores at weeks 8, 12, and 24 (*p* < 0.05). **Conclusions**: Upadacitinib managed to produce lower echography scores much faster than tofacitinib; however, the differences in effectiveness evened out at weeks 12 and 24, with all patients being adequately controlled.

## 1. Introduction

Rheumatoid arthritis (RA) is a chronic, progressive autoimmune disease characterized by synovial proliferation and the destruction of articular cartilage and bone [1,2]. The disability of the patients significantly impairs their quality of life [2,3]. The main goal of treatment is to achieve remission. Otherwise, if the first goal is not possible, the goal is to achieve minimal disease activity [4,5,6,7]. The advent of biologic therapy over the past 25 years has led to significant advances in the treatment of patients with rheumatoid arthritis [8,9]. In addition, the introduction of new small molecules—inhibitors of Janus kinases (JAKs)—offers a new alternative for the treatment of inflammatory joint diseases [10,11,12,13,14,15,16,17,18]. In recent years, the results of numerous clinical trials have proven their effectiveness [10,11,12,13,14,15,16,17,18]. However, there is still a lack of sufficient evidence from real clinical practice to confirm and validate the good therapeutic response and higher remission rate in patients with RA [19,20].

Disease states may be assessed by several scores. The disease activity score (DAS28) evaluates disease severity based on the assessment of 28 joints [21,22]. The clinical disease activity index (CDAI) and the simplified disease activity index (SDAI) are the other two scores frequently used in rheumatology practice [22,23]. Several publications have shown that treatment with tofacitinib and upadacitinib significantly reduces the disease activity assessed by DAS28–ESR and DAS28-CRP [15,18,24]. Moreover, recent evidence has demonstrated the superior effect of upadacitinib in combination with methotrexate (MTX) versus adalimumab + MTX in the inhibition of structural damage [20]. However, these simplified measures cannot compare to ultrasound evaluation, which can assess the reduction in disease activity in terms of radiographical progression. Therefore, full evaluation of the remission state remains questionable without the use of musculoskeletal ultrasound.

In 2017, the European Alliance of Associations for Rheumatology (EULAR) published recommendations for the use of musculoskeletal ultrasound in the management of RA [25]. The new ultrasound definitions (US) and quantifications of synovial hypertrophy (SH) and power Doppler (PD) signal, separately and in combination with the European League Against Rheumatism–Outcomes Measures in Rheumatology (EULAR-OMERACT) combined score for PD and SH, demonstrated moderate–good reliability of ultrasound in RA [25]. Images in gray-scale can provide additional information to regular joint assessment. PD signal is an extremely sensitive marker for joint and tendon inflammation, which can supplement the evaluation of therapeutic response [26]. As Di Matteo et al. pointed out, US is applicable from the prediction of the progression of RA to the confirmation of early diagnosis during the disease continuum [27].

Clinicians are still working on clarifying the principal US findings in RA, scoring systems for evaluation, and appropriate use of US in areas of RA diagnosis and disease prognostication [28,29]. Thus, musculoskeletal ultrasound is becoming a reliable method for the assessment of disease activity, not only in routine clinical practice but also in clinical trials [25,30].

In this pilot study, we aimed to evaluate the ultrasound response to treatment with tofacitinib and upadacitinib on tendons and joints in comparison to clinical and laboratory results in patients with RA who have shown inadequate response to conventional synthetic and/or biologic disease-modifying antirheumatic drugs (cs/b DMARDs). In addition, we performed a musculoskeletal ultrasound at each visit to assess early response and maintenance of the therapeutic outcome during the 24-week follow-up period.

## 2. Materials and Methods

### 2.1. Design of the Study

This was a prospective, observational, longitudinal study at the largest rheumatology clinic at the University Hospital “St. Ivan Rilski” in Sofa, Bulgaria, during the period 2018–2022. It was conducted in accordance with the 1989 Declaration of Helsinki and was approved by the Local Ethics Committee (Ethics Approval Protocol Number: 02/06.03.2018). All of the 53 patients provided written informed consent at the time of the first visit, out of which 47 remained at the end of the observation—week 24. The treatment with oral tofacitinib (TOF) or upadacitinib (UPA) was prescribed according to local guidelines and National Health Insurance Fund (NHIF) requirements for disease activity [31,32].

The rationale behind the selection of tofacitinib and upadacitinib was based on the study of Fleischmann RM et al. [20] (Figure 1). Janus kinase inhibitors (JAKi) are relatively new molecules in the armamentarium of RA therapy, and they are especially newly reimbursed in our country, which provoked our interest.

The inclusion criteria for this study were:

Age > 18 years < 85 years.

Clinically proven RA according to ACR (1987) and/or ACR/EULAR (2010) criteria.

Different disease duration.

Patients treated in combination with or without methotrexate or another csDMARD in a stable dose, and/or corticosteroid (CS) therapy up to 10 mg/daily, and/or non-steroidal anti-inflammatory drugs (NSAIDs).

Patients treated with bDMARD (up to 2 bDMARDs, discontinued 3 months before initiation of the treatment with JAK-inhibitor).

Patients in the individual treatment groups did not change the dosage regimen or discontinue csDMARD or CS therapy during the entire follow-up period.

NSAIDs should be at a stable dose throughout the study period.

The exclusion criteria were defined as presence of infectious diseases, cardiac insufficiency (NYHA III and IV grade), malignant hypertension, any neoplasms, or proliferative lymph diseases within the previous 5 years.

All patients were followed up for a period of 24 weeks through six visits—baseline, week 2, week 4, week 8, week 12, and week 24 after the initiation of tofacitinib or upadacitinib. Joint assessment was performed during each visit by the same rheumatologist. Blood samples were collected to measure the levels of C-reactive protein. Laboratory tests comprised hematology, liver and kidney function, lipid profile, HBV, HCV screening, interferon-gamma release assay for latent tuberculosis (LTB), rheumatoid factor (RF), and anti-cyclic citrullinated peptide antibody (ACPA), which were obtained at baseline. Radiographies of hands and the chest were performed within 6 months before enrollment. We evaluated the disease activity using DAS28-CRP [21]. In addition, we measured CDAI and SDAI to precisely define the severity of the disease in each patient and searched for correlation with ultrasound scores. The three parameters were calculated during each visit.

DAS28 was calculated as follows:DAS28 (CRP) = 0.56*√(TJC28) + 0.28*√(SJC28) + 0.014*GH + 0.36*ln(CRP + 1) + 0.96
where

TJC = tender joint count, and SJC = swollen joint count.

CDAI was calculated as follows:
CDAI = SJC(28) + TJC(28) + PGA + EGA,
where

SJC(28) = swollen 28-joint count; TJC(28) = tender 28-joint count; PGA = patient global disease activity; and EGA = evaluator’s global disease activity.

SDAI was calculated as follows:SDAI = SJC + TJC + PGA + EGA + CRP
where SJC = swollen joint count; TJC = tender joint count; PGA = patient global assessment of disease activity; EGA = evaluator global assessment of disease activity; and CRP = CRP in mg/dL.

### 2.2. Musculoskeletal Ultrasound

Ultrasound evaluation was performed using Esaote model ultrasound machines, MyLab Twice, with a 4–15 MHz frequency probe. Power Doppler parameters were adjusted with a pulse repetition rate ranging between 400 and 800 Hz. Two independent sonographers performed ultrasound examinations in a darkened room in accordance with standardized scans. Each patient was assessed for two joint scores—gray-scale (GS) and PD (USGS-J and USPD-J); the semi-quantitative scale was calculated from 0 to 3 for each joint, and a different score was calculated according to the OMERACT-EULAR composite US scale (OMERACT) from 0 to 3 for each joint. In addition, we assessed the tenosynovitis and calculated USGS-T and USPD-T according to the OMERACT-EULAR scoring system [25]. The targeted ultrasound initiative (TUI) synovitis evaluation form was used to record the results.

### 2.3. Statistical Analysis

Data were collected and analyzed in Microsoft Office’s Excel, with additional descriptive statistics performed by MedCalc 21.2 statistical software. Standard deviations of average scores are included in Table 1 and Table 2. The Pearson correlation coefficient was calculated for echographic variables vs. disease activity variables. A two-sample *t*-test was applied to mean values and obtained for all weeks of measurement, comparing upadacitinib vs. tofacitinib. Correlation matrixes were constructed through R studio, both separately for each drug and using a combined matrix to better follow changes. The main correlation coefficients are present on the graphs with the significance level in Tables 4 and 5. Baseline scores were available only for ultrasound measures and are presented additionally in Table 2 and Figure A1.

## 3. Results

### 3.1. Patient Population

A total of 53 patients were recruited, with 6 patients lost to follow-up due to adverse drug reactions (ADRs) or due to therapy changes. A total of 25 patients were on upadacitinib therapy and 22 on tofacitinib. The average age was 56 years old, with the majority of patients being women. The average durations of disease were 8.24 and 11.17 years, respectively (Table 1).

### 3.2. Disease Activity Scores

Baseline scores for all ultrasound measures and both groups of patients showed similarity (Table 2).

Both groups of patients experienced a steady decline in symptom severity and disease activity, as measured by SDAI, CDAI, and DAS-28. A steady linear downward trend was observed from week 2 to week 24. Patients on upadacitinib had average DAS-28 scores of 5.59 at week 2 and 2.93 at week 24 vs. 4.87 and 2.85 for tofacitinib (Table 3).

CDAI and SDAI scores decreased dramatically between weeks 2 and 4. No significant differences between the scores for both groups were observed, apart from week 2, where patients on tofacitinib had lower average scores for CDAI and SDAI (Figure 2 and Figure 3)

### 3.3. Echography Scores

Both medicinal products managed to decrease the echography score in all domains. Tofacitinib managed to produce a linear reduction in echography scores, with the average lowering by roughly the same amount upon each successive measurement (Figure 4).

Upadacitinib managed to produce lower echography scores much faster than tofacitinib; however, the differences in effectiveness evened out at weeks 12 and 24, with all patients being adequately controlled (Figure 5). Of particular note is the fact that patients on upadacitinib showed a marked improvement beginning from week 2, with USGS-J experiencing a 10-point average reduction from baseline; USPD-J a 7.2-point reduction; OMERACT composite a 10.7-point reduction; USGS-T a 3.4-point reduction; and USPD-T a 2.6-point reduction.

### 3.4. Correlation Analysis

Table 4 and Table 5 show where the correlation measured reached statistical significance in relation to echography scores. Overall, there was a correlation between echography scores and disease activity measurements. Patients with low DAS 28, CDAI, and SDAI also showed lower joint and tendon inflammation. DAS 28 was the most sensitive to this correlation.

Despite the average echography being low from week 2 for upadacitinib, this was not reflected as much in the correlation analysis. A significant correlation began to be observed from week 8 onward.

Contrary to that, the linear reduction for tofacitinib resulted in higher correlation coefficients, as well as statistically significant correlations beginning from week 2. Here, both DAS 28-CRP and CDAI were sensitive enough to confirm this correlation.

Overall, results showed that both products led to a measurable clinical response while at the same time reducing inflammation in the joints and tendons. Weeks 12 and 24 showed the highest correlation coefficients, which was to be expected, since both products achieved disease control at a similar rate from week 8 onward.

As DAS-28-CRP was the most sensitive to measuring correlation, the following graphs show how the trendlines shifted for both products. Tofacitinib results were more in line with expected correlation coefficients; however, there seemed to be a mismatch for upadacitinib. That is to say, despite echography scores being low from week 2 onward, it seemed clinical disease activity measures were able to capture this beginning from week 8 onward. Figure 6 captures this gradual progression for USGS-J as a point illustration.

Although tofacitinib trends appeared to be more predictable, both sets of measurements showed a clear trend—lower echography scores corresponded to lower disease state scores, which appeared to be significant. Correlation matrices were constructed through R studio, both separately for each drug and as a combined matrix to better follow the changes (Appendix B).

## 4. Discussion

Imaging techniques have long been used in rheumatoid arthritis and is one of the main sources of understanding the pathology of the disease, helping physicians assess both synovial and bone damage. Better disease understanding, treat-to-target approaches, advent of TNF-alpha inhibitors, and better csDMARD usage have allowed patients to experience significant improvement in disease course [33]. This includes better outcomes, higher remission rates, less radiographical damage, and better quality of life [34]. Despite these improvements, many patients still fail to respond to therapy. JAK inhibitors, small-sized oral molecules, have provided a targeted approach to treat these patients, with similar efficacy demonstrated by clinical trials [35]. However, these molecules are still accumulating evidence in real-world settings, and our study adds to the body of evidence, showing results consistent with randomized clinical trials (RCTs) [36,37,38,39].

Progression of joint damage is linked with disease activity, with studies showing that TNF-alpha inhibitors disrupt this link [40]. Assessing disease activity and extensive radiography evaluation is not commonplace, with studies of tofacitinib assessing only the van der Heijde Total Sharp Score [41], while upadacitinib studies have also assessed erosion scores and joint space narrowing as measurements [42]. In 2008, Hameed et al. found that composite US markers of synovial disease relate significantly to DAS28 [43]. The study compared RA patients with normal controls only in terms of imaging and disease severity, without investigating the impact of therapy. Our findings corroborate the results of that study, showing that DAS28 is extremely sensitive to ultrasound improvement while additionally providing point estimates.

Our study showed that upadacitinib achieved rapid effect in the first 2–4 weeks, which gradually leveled out over the 6-month observation period, leading to stability in disease management. Similarly, Baldi et al. followed upadacitinib patients for 6 months in three Italian rheumatology centers, observing DAS28-CRP, SDAI, CDAI, USGS, and USPD improvements in the first month [44]. However, in that study, patients were examined at months 1, 3, and 6, showing gradual improvement. The additional measurement of patients at week 2 showed that the effect was more rapid than previously suspected. This has implications for clinical decision making when patients with severe RA require quick, clinically meaningful disease control. The other study for upadacitinib reported that after 12 weeks and 24 weeks, 40% and 63.6% of patients, respectively, achieved US plus clinical remission [45].

Our results confirmed the data from the aforementioned studies, providing additional information about the change in disease activity in patients between weeks 2, 4, 8, and 12.

The improvement with tofacitinib was non-inferior to that of upadacitinib, with patients also experiencing improvement as early as week 2 of treatment, continuing up to month 6. However, our findings showed that this decrease was more linear and gradual, with a high correlation to all disease activity measurement scales, observable from week 2 as well. The findings of our ultrasound measurements are in line with what Germano et al. discovered in a real-world study with an identical study design [19], where joint and tendon scores were also significantly reduced at week 2. Curiously, as the authors themselves noted, they did not find any correlation between the variations of DAS28-CRP and any US scores at any visits. Our findings dispute this claim, showing that the gradual control offered by TOF is linked with improvement in DAS28 scores and seem to be more in line with other studies in the field, such as Razmjou et al. [26], whose study showed a correlation between US gray-scale and power Doppler parameters from baseline to week 2 and a persistent correlation with changes in CDAI and DAS28. Similarly, Ceccarelli et al. [46] found significant correlation between changes in DAS28-CRP and changes in the mean synovial hypertrophy score for both joints with and without Doppler activity. These results revealed that the improvement of the joints is directly related to the reduction of disease activity and the overall improvement of the condition of the patients.

The implementation of the EULAR/OMERACT scoring system in our study assessed joint inflammation using the combination of gray-scale and color power Doppler. The addition of the OMERACT composite score in combination with all of the other routine ultrasound scores offered a more precise evaluation of the regression of the joint inflammatory process. Our results are very similar to the very small number of studies from real-life clinical practice that evaluate the correlations between clinical parameters and US scores in patients with RA treated with tofacitinib [19]. However, one of the strengths of our study is the direct head-to-head comparison of disease activity assessed by ultrasound and DAS28-CRP, CDAI, and SDAI in patients treated with tofacitinib and upadacitinib in a real-life clinical setting. In addition, it presents the important role of ultrasound in the evaluation of remission and low disease activity and reveals discrepancies between the results reported by DAS28-CRP, CDAI, and SDAI. Another strength of our study is the frequent MSUS assessment of the patients (baseline and weeks 2, 4, 8, 12,16, and 24) and the results that highlight the early effectiveness of the treatment with both tsDMARDs.

The limitations of our study are the single-center participation, sample size of 47 patients, and open-label design, as well as the initial evaluation of disease activity. Some of these limitations are overcome by the fact that this is the reference rheumatology clinic for Bulgaria and admits a variety of complicated cases, so our sample includes difficult-to-treat patients. Furthermore, the collection of ultrasound measures for all 47 patients is a very intensive resource undertaking, so we can consider this study as the pilot for our hospital. Further enlargement of samples needs to be performed when the JAKi starts to be more frequently prescribed, and we can recruit more patients.

## 5. Conclusions

In conclusion, the results obtained provide evidence that treatment with both tofacitinib and upadacitinib leads to an early clinical response to treatment (week 2) and a long-lasting reduction in US signs of inflammation. However, further investigations are needed to make general conclusions about which JAK inhibitor has an early effect and whether the selectivity of the mechanism of action has a crucial role in determining the better effect on the signs of inflammation. Our study highlights the individual nature of patient responses to therapy.

## Figures and Tables

**Figure 1 biomedicines-13-01339-f001:**
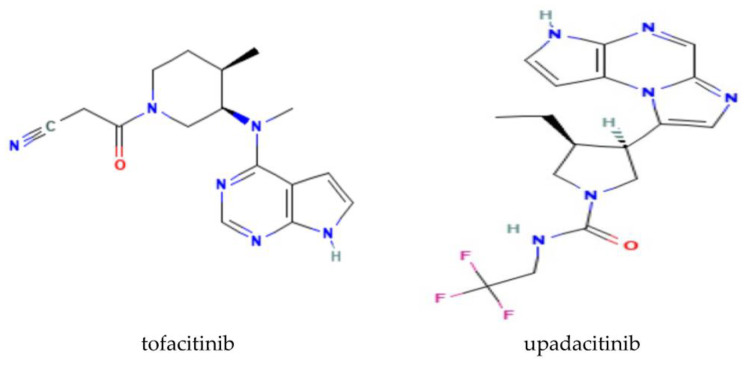
Structural 2D formulas of tofacitinib and upadacitinib.

**Figure 2 biomedicines-13-01339-f002:**
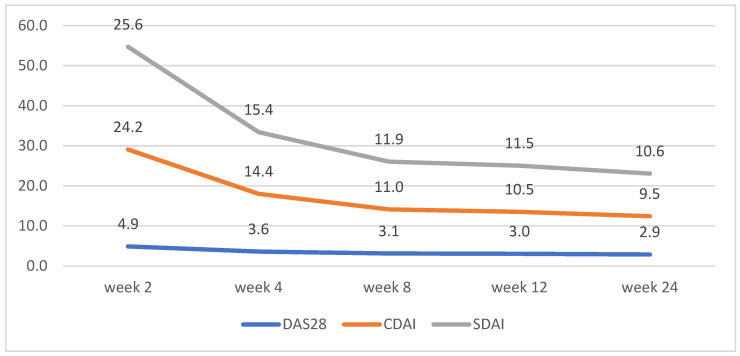
Tofacitinib disease activity trends according to changes in DAS28, CDAI, and SDAI.

**Figure 3 biomedicines-13-01339-f003:**
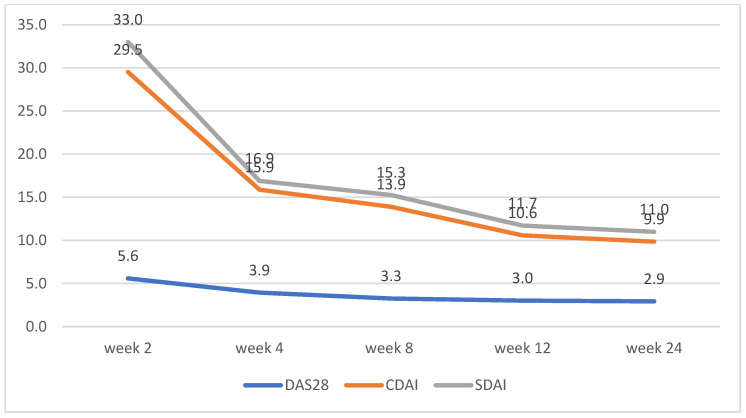
Upadacitinib disease activity trends according to changes in DAS28, CDAI, and SDAI.

**Figure 4 biomedicines-13-01339-f004:**
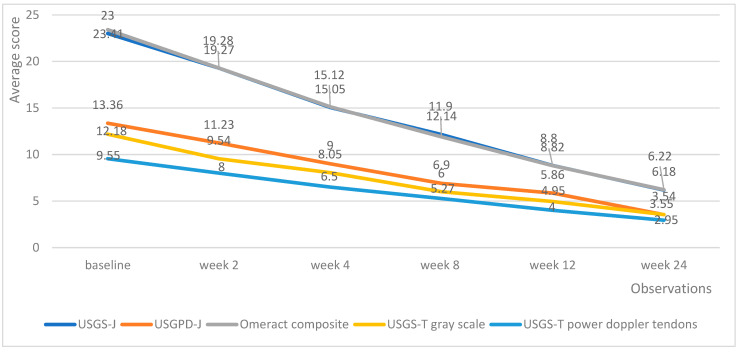
Tofacitinib trends in echography average scores per week.

**Figure 5 biomedicines-13-01339-f005:**
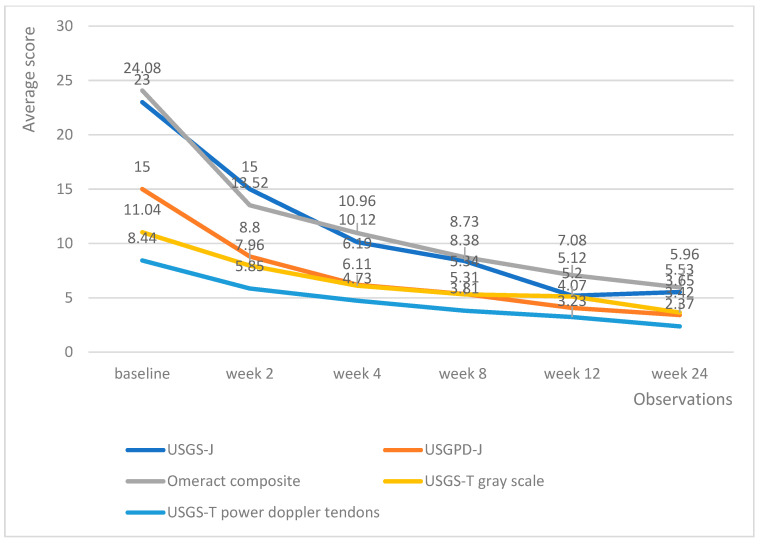
Upadacitinib trends in echography average scores per week.

**Figure 6 biomedicines-13-01339-f006:**
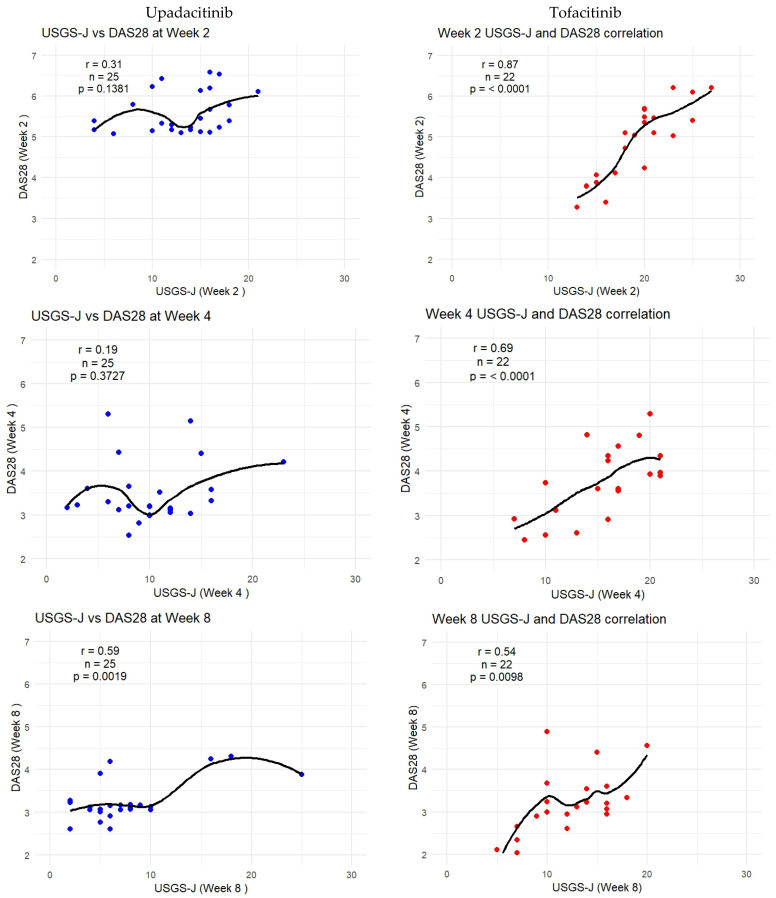
USGS-J score correlations with DAS28-CRP for weeks 2, 4, 8, 12, and 24.

**Table 1 biomedicines-13-01339-t001:** Demographic characteristics of the patients on upadacitinib and tofacitinib.

Demographic Characteristics	Upadacitinib	Tofacitinib
Male	5	4
Female	20	18
Age	56 ± 11.6 SD	56.9 ± 11.3 SD
Duration of disease	8.24 ± 4.92 SD	11.17 ± 7.5 SD
Total Number of patients	25	22

**Table 2 biomedicines-13-01339-t002:** Baseline scores for ultrasonographic (US) measures for both medicines.

	OMERACT Composite		USGS-J		USPD-T		USPD-J		USGT-T	
Drug	tofa	upa	tofa	Upa	tofa	upa	tofa	upa	tofa	upa
N	22	25	22	25	22	25	22	25	22	25
Mean	23.4	24	23	23	9.54	8.44	13.36	15	12.18	11.04
95% CI	21.2–25.5	22.4–25.7	21.0–24.9	21.4–24.5	8.4–10.59	7.4–9.4	12.4–14.2	13.7–16.2	11.1–13.2	9.88–12.1
Variances	23.01	16.57	20.19	13.33	5.59	5.92	4.05	9.00	5.67	7.87
SD	4.7	4.07	4.49	3.65	2.36	2.43	2.01	3.0	2.38	2.80
Median	23.50	24.00	23.0	23.0	10.0	8.000	13.50	16.00	12.0	11.00
95% CI	19.9–26.0	22.0–25.8	19.9–25.04	21.0–25.0	8.00–10.05	7.00–9.00	12.00–15.0	13.0–17.0	10.0–14.0	10.0–12.86
Minimum	16.00	18.00	16.00	16.0	5.000	4.0	10.0	8.000	8.000	6.000
Maximum	35.0	32.0	33.0	30.0	14.00	17.00	16.0	20.0	16.0	16.0
25–75p	19–26	21–27.2	19–26	20.7–26	8–11	7–9	12–15	12.7–17	10–14	8.7–13

Legend: Tofa (tofacitinib); upa (upadacitinib).

**Table 3 biomedicines-13-01339-t003:** Average disease activity scores (DAS28-CRP) during the follow-up period for both medicines.

		DAS28			
Upadacitinib	WEEK 2	WEEK 4	WEEK 8	WEEK 12	WEEK 24
Average	5.5932	3.4904	3.2552	3.0152	2.93
Standard deviation	0.5081	0.4805	0.4734	0.5344	0.759
Tofacitinib	WEEK 2	WEEK 4	WEEK 8	WEEK 12	WEEK 24
Average	4.8705	3.5964	3.1191	3.0345	2.8591
** *Standard deviation* **	** *0.9134* **	** *0.9453* **	** *0.8357* **	** *0.7866* **	** *0.6574* **
		**CDAI**			
Upadacitinib	WEEK 2	WEEK 4	WEEK 8	WEEK 12	WEEK 24
Average	29.52	15.884	13.8956	10.5784	9.8536
Standard deviation	5.7783	6.2021	4.8695	5.6715	7.1093
Tofacitinib	WEEK 2	WEEK 4	WEEK 8	WEEK 12	WEEK 24
Average	24.1864	14.3818	10.9864	10.4773	9.5473
** *Standard deviation* **	** *5.7063* **	** *7.1046* **	** *5.528* **	** *4.7828* **	** *3.9626* **
		**SDAI**			
Upadacitinib	WEEK 2	WEEK 4	WEEK 8	WEEK 12	WEEK 24
Average	33.0064	16.8944	15.264	11.72	10.9952
Standard deviation	6.6384	6.5034	5.6479	5.93	7.4371
Tofacitinib	WEEK 2	WEEK 4	WEEK 8	WEEK 12	WEEK 24
Average	25.6241	15.4141	11.93	11.5109	10.6455
** *Standard deviation* **	** *6.1324* **	** *7.753* **	** *6.1275* **	** *5.3767* **	** *3.7905* **

**Table 4 biomedicines-13-01339-t004:** Statistically significant correlations between echography and disease activity scores of upadacitinib.

		USGS-J	USPD-J	OMERACT Composite	USGS-T	USPD-T
DAS28	Week 2	No	No	No	No	No
	Week 4	No	No	No	No	No
	Week 8	Yes *p* = 0.0019	Yes *p* = 0.0049	Yes *p* = 0.0022	Yes *p* = 0.0010	Yes *p* = 0.0007
	Week 12	Yes *p* = 0.0013	Yes *p* = 0.0010	Yes *p* = 0.0017	Yes *p* = 0.0002	Yes *p* = 0.0004
	Week 24	Yes *p* = 0.0180	Yes *p* = 0.0241	Yes *p* = 0.0321	Yes *p* = 0.0145	Yes *p* = 0.0174
CDAI	Week 2	No	No	No	No	No
	Week 4	No	No	No	No	No
	Week 8	Yes *p* = 0.0443	No	No	Yes *p* = 0.0173	Yes *p* = 0.0273
	Week 12	Yes *p* = 0.0249	Yes *p* = 0.0314	Yes *p* = 0.0282	Yes *p* = 0.0188	Yes *p* = 0.0146
	Week 24	Yes *p* = 0.0151	Yes *p* = 0.0151	Yes *p* = 0.0154	Yes *p* = 0.0170	Yes *p* = 0.0189
SDAI	Week 2	No	No	No	No	No
	Week 4	No	No	No	No	No
	Week 8	No	No	No	Yes *p* = 0.0280	No
	Week 12	Yes *p* = 0.0135	Yes *p* = 0.0175	Yes *p* = 0.0151	Yes *p* = 0.0101	Yes *p* = 0.0075
	Week 24	Yes *p* = 0.0084	Yes *p* = 0.0099	Yes *p* = 0.0105	Yes *p* = 0.0078	Yes *p* = 0.0075

**Table 5 biomedicines-13-01339-t005:** Statistically significant correlations between echography and disease activity scores of tofacitinib.

		USGS-J	USPD-J	OMERACT Composite	USGS-T	USPD-T
DAS28	Week 2	Yes *p* < 0.0001	Yes *p* = 0.0126	Yes *p* < 0.0001	Yes *p* = 0.0027	Yes *p* = 0.0167
	Week 4	Yes *p* = 0.0004	Yes *p* = 0.0447	Yes *p* = 0.0003	Yes *p* = 0.0466	Yes *p* = 0.0293
	Week 8	Yes *p* = 0.0098	No	Yes *p* = 0.0146	Yes *p* = 0.0535	No
	Week 12	Yes *p* = 0.0081	Yes *p* = 0.0113	Yes *p* = 0.0089	Yes *p* = 0.0041	Yes *p* = 0.0372
	Week 24	Yes *p* = 0.0001	Yes *p* = 0.0004	Yes *p* = 0.0001	Yes *p* < 0.0001	Yes *p* < 0.0001
CDAI	Week 2	Yes *p* = 0.0002	No	Yes *p* = 0.0002	Yes *p* = 0.0163	No
	Week 4	Yes *p* = 0.0052	No	Yes *p* = 0.0032	No	No
	Week 8	No	No	No	Yes *p* = 0.0468	No
	Week 12	Yes *p* = 0.0093	Yes *p* = 0.0242	Yes *p* = 0.0182	Yes *p* = 0.0065	Yes *p* = 0.0114
	Week 24	Yes *p* = 0.0018	Yes *p* = 0.0086	Yes *p* = 0.0024	Yes *p* = 0.0007	Yes *p* = 0.0005
SDAI	Week 2	Yes *p* = 0.0001	No	Yes *p* = 0.0001	Yes *p* = 0.0056	Yes *p* = 0.0296
	Week 4	Yes *p* = 0.0033	No	Yes *p* = 0.0021	Yes *p* = 0.0468	No
	Week 8	No	No	No	No	No
	Week 12	Yes *p* = 0.0104	Yes *p* = 0.0266	Yes *p* = 0.0130	Yes *p* = 0.0032	No
	Week 24	Yes *p* = 0.0012	Yes *p* = 0.0083	Yes *p* = 0.0014	Yes *p* = 0.0003	Yes *p* = 0.0007

## Data Availability

Data are unavailable due to privacy or ethical restrictions.

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
