# Peer review of "Ultrasound Evaluation of Therapeutic Response to Tofacitinib and Upadacitinib in Patients with Rheumatoid Arthritis—Real-Life Clinical Data"

_biomedicines, 2025, doi:10.3390/biomedicines13061339_

Round 1
Reviewer 1 Report (Previous Reviewer 1)
Comments and Suggestions for Authors
The authors have made significant changes. But some corrections are still required
- The figures are not of good quality in terms of resolution and presentation. It may not eventually qualify the journal standards. In figures, the values should be given differently, like instead of 24.1864, replace it with 24.2, and so on. In many figures, the values are overlapping. Figure 3.4,5 needs better representation. Hardly anything is readable
- The authors must elaborate on the information from Ref 20, which talks about adalimumab. Is there any clinical trial where Adalimumab is used alone or in combination with JAKi, other kinase inhibitors, or Ultrasound for RA? Please provide that reference and compare it with your data in the discussion and conclusion.
- Show the structure of tofacitinib and Upadacitinib somewhere in the manuscript.
Author Response
Dear Reviewers,
We revised the manuscript in accordance with yours’ recommendations. The changes and corrections in the revised version of our manuscript are marked in red color in the text.
We also responded to each reviewer’s queries in a point – by – point manner. The revised version of the manuscript and the replies to the reviewers have been submitted online.
Thank you for your time and efforts.
Responses to the Reviewers
Reviewer 1
1. The figures are not of good quality in terms of resolution and presentation. It may not eventually qualify the journal standards. In figures, the values should be given differently, like instead of 24.1864, replace it with 24.2, and so on. In many figures, the values are overlapping. Figure 3.4,5 needs better representation. Hardly anything is readable
Thank you for this recommendation. All of the figures are corrected and the values are changed.
2. The authors must elaborate on the information from Ref 20, which talks about adalimumab. Is there any clinical trial where Adalimumab is used alone or in combination with JAKi, other kinase inhibitors, or Ultrasound for RA? Please provide that reference and compare it with your data in the discussion and conclusion.
Additional studies are included in discussion part. Ref. N: 45. A clinical trial with the same drug is not mentioned/discussed in the article: Picchianti Diamanti, A., Cattaruzza, M.S., Salemi, S. et al.Clinical and Ultrasonographic Remission in Bio-naïve and Bio-failure Patients with Rheumatoid Arthritis at 24 Weeks of Upadacitinib Treatment: The UPARAREMUS Real-Life Study. Rheumatol Ther 11, 1347–1361 (2024). https://doi.org/10.1007/s40744-024-00712-y
3. Show the structure of tofacitinib and Upadacitinib somewhere in the manuscript.
We include new Figure 1.
Reviewer 2 Report (New Reviewer)
Comments and Suggestions for Authors
Dear Authors,
The article “Ultrasound evaluation of therapeutic response to tofacitinib and upadacitinib in patients with rheumatoid arthritis – real life clinical data” introduces prospective study in which patients were treated with upadacitinib or tofacitinib. The objective of the study is actual and the results are important for clinicians.
There are some omissions, which are listed below.
- The name of the tables and figures should contain full information. The reader must have possibility to explore them independently on the text.
- Figure 3 – two digits after dot is enough
- It is hard to read Figure 4. Please, make it clearer.
- What does the symbol ” – “ mean in the Table 4, 5 for the SDAI and CDAI 2,4,8 weeks? It is desirable to give a Note under the tables and additionally disclose abbreviations.
- t is written in the abstract “Disease activity was assessed by DAS28-CRP, CDAI, SDAI, as well as MSUS.” There is a lot of ABB in the text. Besides, DAS28 – ESR and DAS28-CRP is not abbreviated.
- The reference is needed for: The treatment with oral Tofacitinib (TOF) or Upadacitinib (UPA) was prescribed according to local guidelines of the National Health Insurance Fund (NHIF) requirements for disease activity. In any way it is desirable to explain the choice of these drugs.
- A clinical trial with the same drug is not mentioned/discussed in the article: Picchianti Diamanti, A., Cattaruzza, M.S., Salemi, S. et al.Clinical and Ultrasonographic Remission in Bio-naïve and Bio-failure Patients with Rheumatoid Arthritis at 24 Weeks of Upadacitinib Treatment: The UPARAREMUS Real-Life Study. Rheumatol Ther 11, 1347–1361 (2024). https://doi.org/10.1007/s40744-024-00712-y
- The reference [44] is absent in the text
Author Response
Dear Reviewers,
We revised the manuscript in accordance with yours’ recommendations. The changes and corrections in the revised version of our manuscript are marked in red color in the text.
We also responded to each reviewer’s queries in a point – by – point manner. The revised version of the manuscript and the replies to the reviewers have been submitted online.
Thank you for your time and efforts.
Responses to the Reviewers
Reviewer 2.
- The name of the tables and figures should contain full information. The reader must have possibility to explore them independently on the text.
Corrected for all tables and figures.
- Figure 3 – two digits after dot is enough
Corrected
- It is hard to read Figure 4. Please, make it clearer.
Corrected completely
- What does the symbol ” – “ mean in the Table 4, 5 for the SDAI and CDAI 2,4,8 weeks? It is desirable to give a Note under the tables and additionally disclose abbreviations.
Corrected- replaced with no and changes the title of the tables as:
- It is written in the abstract “Disease activity was assessed by DAS28-CRP, CDAI, SDAI, as well as MSUS.” There is a lot of ABB in the text. Besides, DAS28 – ESR and DAS28-CRP is not abbreviated.
Corrected – we added legend in the beginning for all abbreviations used in the manuscript
- The reference is needed for: The treatment with oral Tofacitinib (TOF) or Upadacitinib (UPA) was prescribed according to local guidelines of the National Health Insurance Fund (NHIF) requirements for disease activity. In any way it is desirable to explain the choice of these drugs.
Added references for Guideline of rheumatology diseases and NHIF webpage
- A clinical trial with the same drug is not mentioned/discussed in the article: Picchianti Diamanti, A., Cattaruzza, M.S., Salemi, S. et al.Clinical and Ultrasonographic Remission in Bio-naïve and Bio-failure Patients with Rheumatoid Arthritis at 24 Weeks of Upadacitinib Treatment: The UPARAREMUS Real-Life Study. Rheumatol Ther11, 1347–1361 (2024). https://doi.org/10.1007/s40744-024-00712-y
Corrected – added in the discussion section and cited under number 45.
- The reference [44] is absent in the text
Corrected – dismissed and changed references number. In general all references numbering were changed due to addition of new once.
Round 2
Reviewer 2 Report (New Reviewer)
Comments and Suggestions for Authors
Dear Authors,
The article has been improved
Author Response
Thank you very much for your recommendations.
This manuscript is a resubmission of an earlier submission. The following is a list of the peer review reports and author responses from that submission.
Round 1
Reviewer 1 Report
Comments and Suggestions for Authors
Reviewer’s comments for biomedicines-3048074.
1. The authors have done an observational study on 47 patients. The sample size is too low to give a conclusion or compare with other existing data considering that it is only an observational study. Further, there are too many abbreviations in the Abstract without giving their full names.
2. The graphs are very poorly presented. In some of the graphs, the line is too thin to be even visible. The data for every score (like DAS etc) should have been presented for both drugs in one graph for a better comparison. Further, instead of Excel other graph-plotting software should be used. Why are so many graphs for correlation coefficients? Again, for both drugs the values should be presented in a single graph.
3. How the DAS, CDAI, and other scores are calculated? what are the parameters?
4. In lines 103-106 authors are talking about several tests. Have they taken the data of these tests from the clinical set-up? If so, how these data are presented and where?
5. Why there is no standard deviation (SD) in any of the graphs?
6. why the scores of Week 0 is not presented when the patients are recruited for this therapy? Do they have any significant difference in scores between males and females?
7. Adalimumab is a well-known treatment for RA. However, the authors claim that these two drugs and Ultrasound treatment are better. Based on which data? do they have any observational data for a similar setup under adalimumab treatment? If not, this conclusion is NOT valid.
8. although the introduction and conclusion are well written I believe this data set is not good enough for publication in this journal.
Comments on the Quality of English Language
Mostly Okay
Author Response
Dear Reviewer,
Thank you once again for your time.
Attached you will find detailed responses to your comments.

Reviewer 2 Report
Comments and Suggestions for Authors
In this study, musculoskeletal ultrasonography (MSUS) was utilized to assess the therapeutic response of 47 RA patients who were treated with tofacitinib and upadacitinib for 24 weeks, demonstrating that both JAK inhibitors effectively control inflammation. The clinical study excelled in sample collection and follow-up duration, but we sincerely offer the following recommendations:
1. To ensure comparability of MSUS results, it is crucial to use standardized and typical pathological evidence. However, the introduction of musculoskeletal ultrasonography for RA diagnosis in this paper is inadequate. Please provide a comprehensive supplement to enhance the completeness of the article.
2. Clarify the rationale behind the selection of tofacitinib and upadacitinib.
3. The baseline data of the included cohort is insufficient. Additionally, if the studies on tofacitinib and upadacitinib involved two separate cohorts, were confounding factors adequately controlled?
4. While DAS28-CRP was used as the index for evaluating RA disease activity, it is recommended to include additional indicators to enhance the paper's credibility.
Comments on the Quality of English Language
The manuscript is of average English quality and subject to moderate revision.
Author Response
Dear Reviewer,
Thank you once again for your time and efforts.
Attached you will find detailed responses to your comments.
Best regards,
